# Combining visible-light induction and copper catalysis for chemo-selective nitrene transfer for late-stage amination of natural products

Qi Xing [1][✉], Ding Jiang[1], Jiayin Zhang[1], Liangyu Guan[1], Ting Li[1], Yi Zhao[1], Man Di[1], Huangcan Chen[1], Chao Che [1,2][✉] & Zhendong Zhu [1][✉]

Nitrene transfer chemistry is an effective strategy for introducing C–N bonds, which are ubiquitous in pharmaceuticals, agrochemicals and diverse bioactive natural products. The development of chemical methodology that can functionalize unique sites within natural products through nitrene transfer remains a challenge in the field. Herein, we developed copper catalyzed chemoselective allylic C–H amination and catalyst-free visible-light induced aziridination of alkenes through nitrene transfer. In general, both reactions tolerate a wide range of functional groups and occur with predictable regioselectivity. Furthermore, combination of these two methods enable the intermolecular chemo-selective late-stage amination of biologically active natural products, leading to C–H amination or C=C aziridination products in a tunable way. A series of control experiments indicate two-step radical processes were involved in both reaction systems.

[1] BayRay Innovation Center, Shenzhen Bay Laboratory, Shenzhen 518132, China. [2] State Key Laboratory of Chemical Oncogenomics and Key Laboratory of Chemical Genomics, Peking University Shenzhen Graduate School, Shenzhen 518055, China. [✉]email: xingqi@szbl.ac.cn; chec@pku.edu.cn; zhuzd@szbl.ac.cn

**N**atural products play an important role in chemistry and drug discovery[1–3]. In the past few decades, chemists have made enormous efforts to create structurally complex, diverse molecules resembling natural products[4–8]. Since traditional synthesis of natural product derivatives from simple starting materials can be laborious and ineffective, an alternative approach is to derivatize natural products directly *via* selective reactions. With this approach, synthetic routes may be shortened and more effective means can be provided to produce diverse natural products derivatives[9–15]. Since nitrogen-containing molecules are the key structural constituent of many pharmaceutical compounds that play a pivotal role in drug development, the development of methodology for the late-stage introduction of nitrogen-containing functionalities would be valuable to medicinal chemistry[16–20].

Transformations of C–H and C=C bonds to C–N bonds through transition-metal catalyzed nitrene transfer are powerful synthetic tools to prepare valuable amine building blocks[21–41]. Control over the chemoselectivity of nitrene insertion into either C–H or C=C bond was primarily limited to *intramolecular* transformation (Fig. 1a)[42–48]. More recently, it has been shown that the chemoselectivity of *intermolecular* nitrene transfer can be controlled by changing the metal identity or altering the ligands (Fig. 1b)[49–51]. Although several methods have been reported, the chemoselective amination of unsaturated natural products by these reactions has been limited to examples containing a minimum of functional groups. A rare example of selective intermolecular nitrene insertion for functionalizing unique sites within unsaturated natural products employs the rhodium (II)-based system developed by Du Bois[52]. With the rhodium (II) catalyst, the

chemoselectivity for C–H amination and alkene aziridination is highly substrate-dependent. For example, geraniol and parthenin gave site-selective aziridination products, while allylic C–H amination occurred selectively with eupalmerolide and Eupalmerin acetate. So, developing efficient, inexpensive reaction systems for predictable, tunable and intermolecular chemoselective nitrene transfer for late-stage amination of unsaturated natural products remains to be addressed.

Schomaker and co-workers have demonstrated that silver(I) catalysis offers unique opportunities to tune the chemo- and site-selectivity of intramolecular or intermolecular nitrene transfer[53,54]. Copper, which is in the same group as silver, has been frequently employed for aziridination of olefins and presented promising catalytic activity for C–H amination of sp$^2$ C–H bonds, benzylic C–H bonds and inactivated sp$^3$ C–H bonds in saturated hydrocarbons through nitrene transfer[28,55–58]. However, copper-catalyzed selective amination of allylic C–H bonds within unsaturated hydrocarbons through nitrene insertion has been underexplored, probably owing to a preferred reactivity for aziridination with copper catalysis[28,57]. Considering the low cost of copper and its compatibility with biology, developing tunable copper-based catalytic system for selective late-stage C–H amination of unsaturated natural products would be helpful for performing arming or Structure-Activity Relationship (SAR) studies of unsaturated natural products. On the other hand, the selective aziridination of natural products is also of great importance because aziridines are found in diverse biologically active compounds[59]. Also, the ring constraint of aziridines render them valuable intermediates as building blocks for a plethora of chemicals by means of stereo- and regiospecific transformations,

**Fig. 1 Chemo- and site-selective nitrene transfer. a** Intramolecular nitrene transfer reaction of unsaturated hydrocarbon compounds. **b** Intermolecular nitrene transfer reaction of unsaturated hydrocarbon compounds. **c** Combining copper and photo catalysis for nitrene transfer reaction of unsaturated hydrocarbon compounds.

**Table 1 Optimization of the reaction conditions for photo-induced aziridination of alkenes[a].**

| Entry | Change from "standard conditions" | Yield[b] (%) |
|---|---|---|
| 1 | None | 79 |
| 2 | In the dark | n.r. |
| 3 | In air instead of $N_2$ | 72 |
| 4 | PhI=NTs instead of **1a** | 66 |
| 5 | DCE instead of $CH_3CN$ | 53 |
| 6 | THF instead of $CH_3CN$ | n.d. |
| 7 | DMF instead of $CH_3CN$ | n.d. |

[a]Reaction conditions: **1** (0.05 mmol), **2** (0.25 mmol), in $CH_3CN$ (0.5 mL) under 12 W blue light emitting diode (LED) (435–445 nm) at room temperature for 7 h.
[b]Isolated yield.

including ring opening, ring expansion, and rearrangement, and thus lead to more complex nitrogen-containing molecules[60,61]. The transition-metal catalyzed aziridination of alkene with nitrene precursors is the most popular and commonly used approach to construct an aziridine subunit[26,28,34]. Recently, metal-free and catalyst-free aziridination of alkenes have emerged as greener alternatives, but only limited approaches are available[62–65]. Among them, photo-induced aziridination of alkenes with iminoiodinanes seems to be a promising tool, but high energy ultraviolet (UV)-irradiation was used in this protocol[64]. An ideal approach is to develop visible-light induced selective aziridination of alkenes in the absence of any catalysts. Herein, we developed copper catalyzed chemoselective allylic C–H amination and catalyst-free visible light-induced aziridination of alkenes through nitrene transfer (Fig. 1c). Furthermore, combination of these two methods enable the intermolecular chemoselective late-stage amination of biologically active natural products, leading to C–H amination or C=C aziridination products in a tunable way.

## Results and discussion

**Reaction optimization.** Inspired by the rapidly developed photochemistry, we initially investigated the feasibility of utilizing visible light as an energy source for selective aziridination of alkenes. Iminoiodinanes stabilized by *ortho*-substituents were found to be activated under ultraviolet (UV) light irradiation and were used as nitrene precursor for amination reactions[64]. Nemoto and coworkers demonstrated that the hypervalent iodine could also be activated with light from the apparently non-absorbing region *via* a direct $S_0$-$T_n$ transition, which has been considered a forbidden process[66]. Notably, in some cases the generation of an excited triplet state *via* a direct $S_0$-$T_n$ transition is more efficient than that *via* $S_0$-$S_1$-$T_1$ intersystem crossing. Inspired by this, we envisioned that such *ortho*-substituted iminoiodinanes could be similarly activated under light of a longer wavelength, such as visible light, and acted as a nitrene precursor for the aziridination of alkenes. As shown in Table 1, we studied the reaction of cyclohexene **2** with *ortho*-methoxymethyl iminoiodinanes **1** under blue LED irradiation and indeed obtain the aziridination product in good yield (entry 1). Control reaction confirms that no reaction occurred in the absence of visible light (entry 2). When the reaction was carried out in air, the desired product **3** was obtained in 72% yield, indicating that air is tolerated in this reaction (entry 3). In the absence of ortho substituent, iminoiodinane PhI = NTs also reacted to give the corresponding aziridination product, albeit in a lower yield (entry 4). Utilizing (1,2-dichloroethane) DCE instead of $CH_3CN$ as the solvent led to a lower yield (entry 5). Other polar solvents, such as (tetrahydrofuran) THF and (N, N-dimethylformamide) DMF, were notably less effective, mostly giving a mixture of side products (entries 6–7). Besides blue LED, we also tried to use white LED as the light source and a comparable yield was observed (entry 3 in Supplementary Table 1). We also investigated the impact of substrate ratio and found that reducing the amount of alkene to one or two equivalent both led to a slight decrease in product yield (entries 9 and 10 in Supplementary Table 1). No improvement in yield was observed when the reaction time was prolonged to 12 h. (entry 11 in Supplementary Table 1). These results are described in the Supplementary Results and Discussion (Supplementary Table 1).

For the selective allylic C–H amination, copper was selected as the central metal for catalyst screening in view of its low cost and biological compatibility. As previously shown, catalyst control of the nitrene transfer could be achieved by altering the nature of ligands[53,54]. So, we initially investigated the effects of different ligands (Table 2, Supplementary Table 2). In general, phenanthroline and bipyridine ligands favor aziridination (entries 1–5) and oxazolines favor allylic C–H amination (entries 6–12), except for $L_6$ and $L_{10}$. β-Diketiminate ligand ($L_{13}$ and $L_{14}$) and PyBox ligand ($L_{15}$) prefer the aziridination of C=C bond to deliver aziridines as the major products (entries 13–15). Altering the solvent to DCE enhanced the site selectivity in favor of the allylic C–H amination (entry 16). By contrast, (dimethyl sulfoxide) DMSO, (dimethylacetamide) DMA, MeOH and (N-methyl–2-pyrrolidone) NMP were all inefficient solvents for both nitrene transfer reactions (entries 17–20). Finally, a bidentate oxazoline ligand $L_9$ offers advantages in terms of yield and chemoselectivity using DCE as the solvent (entry 22). Notably, the addition of (4 Å Molecular Sieves) 4ÅMS increased the reaction yield substantially and did not affect the chemoselectivity (entry 23). As shown in the Supplementary Results and Discussion, we also conducted parallel experiments of copper-catalyzed C–H amidation with different metal:ligand ratios, from 1:1 to 1:2.5. Compared with the chemoselectivity in 1:1.5 metal/ligand ratio, the yield and chemoselectivity for C–H amination is poorer when the catalyst system was used in 1:1 metal/ligand ratio (entry 10 in Supplementary Table 3). Further increasing the metal/ligand ratio from 1:1.5 to 1:2 or 1:2.5 improved the overall yield to some extent, but failed to improve the chemoselectivity for C–H amination (entries 11 and 12 in Supplementary Table 3). Slightly lower yields were obtained when the amount of alkene was reduced to one or two equivalents, while the chemoselectivity was unaffected (entries 13 and 14 in Supplementary Table 3).

**Table 2 Optimization of the reaction conditions for copper-catalyzed C–H amidation [a,b,d].**

| Entry | Ligand | Solvent | Yield[b] (%) | 3/4 |
|---|---|---|---|---|
| 1 | L1 | CH3CN | 42 | 2.5/1 |
| 2 | L2 | CH3CN | trace | ----- |
| 3 | L3 | CH3CN | trace | ----- |
| 4 | L4 | CH3CN | 21 | 6/1 |
| 5 | L5 | CH3CN | 23 | 4.8/1 |
| 6 | L6 | CH3CN | 34 | 3.9/1 |
| 7 | L7 | CH3CN | 14 | 1/4 |
| 8 | L8 | CH3CN | 20 | 1/1.5 |
| 9 | L9 | CH3CN | 14 | 0.8/1 |
| 10 | L10 | CH3CN | 47 | 2.9/1 |
| 11 | L11 | CH3CN | 24 | 1/5 |
| 12 | L12 | CH3CN | 36 | 1/8 |
| 13 | L13 | CH3CN | 45 | 8/1 |
| 14 | L14 | CH3CN | 9 | 1/0.8 |
| 15 | L15 | CH3CN | 76 | 4.8/1 |
| 16 | L7 | DCE | 34 | 1/7.5 |
| 17 | L7 | DMSO | n.d. | ----- |
| 18 | L7 | DMA | trace | ----- |
| 19 | L7 | MeOH | n.d. | ----- |
| 20 | L7 | NMP | trace | ----- |
| 21 | L8 | DCE | 38 | 1/8.5 |
| 22 | L9 | DCE | 48 | 1/11 |
| 23[c] | L9 | DCE | 67 | 1/10 |

[a]Reaction conditions: **1** (0.1 mmol), **2** (0.5 mmol), [Cu(OTf)]2·toluene (10 mol%), L (15 mol%), r.t., in solvent (0.9 mL) under $N_2$ for 16 h.
[b]Isolated yield.
[c]4ÅMS (100 mg) was added.
[d]p-Toluenesulfonamide, which was generated from decomposition of the iminoiodinane, was obtained as the major side product.

According to the reactivity of iminoiodinanes reported by Takemoto and coworkers, we prepared another two iminoiodinanes and compared their activities in copper-catalyzed C–H amination (Supplementary Scheme 1). In general, the corresponding C–H amination product was successfully obtained in moderate yields using *ortho*-methoxymethyl or *ortho*-nitro iminoiodinanes of *p*-toluenesulfonamide as nitrogen sources, respectively. Both yield and chemoselectivity for C–H amination with *ortho*-methoxymethyl iminoiodinane was slightly higher than that with *ortho*-nitro iminoiodinane. *ortho*-Nitro iminoiodinane of *ortho*-nitrosulfonamide gave only trace amounts of C–H amination and aziridination products, probably due to its poor solubility in DCE.

**Substrate scope**. With the optimal reaction conditions in hand, we set out to explore the substrate scope of the selective aziridination/allylic C–H amination of various alkenes. Under blue LED irradiation, aliphatic aziridines **3–16** were all smoothly generated in moderate to good yields from the corresponding cyclic, 1,1-disubstituted, terminal and internal olefins (Fig. 2a). Electron-poor alkene and diene are also compatible with the reaction conditions, giving **17** and **18** in 23 and 37% yield, respectively. Styrenes having different substituents on the aromatic rings were all tolerated to afford the corresponding aziridines **19–23** in acceptable yields. The disubstituted aromatic olefin provided *anti*-isomer as the main product (**24**). Importantly, alkyl halides and aryl halides that can undergo further coupling reactions were all well-tolerated (**13** and **21**). Under visible-light irradiation, pinene also underwent aziridination smoothly to give the corresponding products in 66% yield, with **25a** as the major isomer. α-Terpinylacetate gives exclusively selective aziridination product with a *anti/syn* ratio of 3.8:1 (**26a** and **26b**). Only aziridination of limonene is observed with photo induction, with the endocyclic alkene preferred slightly over the exocyclic alkene by a ratio of 1:0.8 (**27a** and **27b**).

Then olefins containing both C=C and C–H bonds were selected to examine the robustness of copper-catalyzed allylic

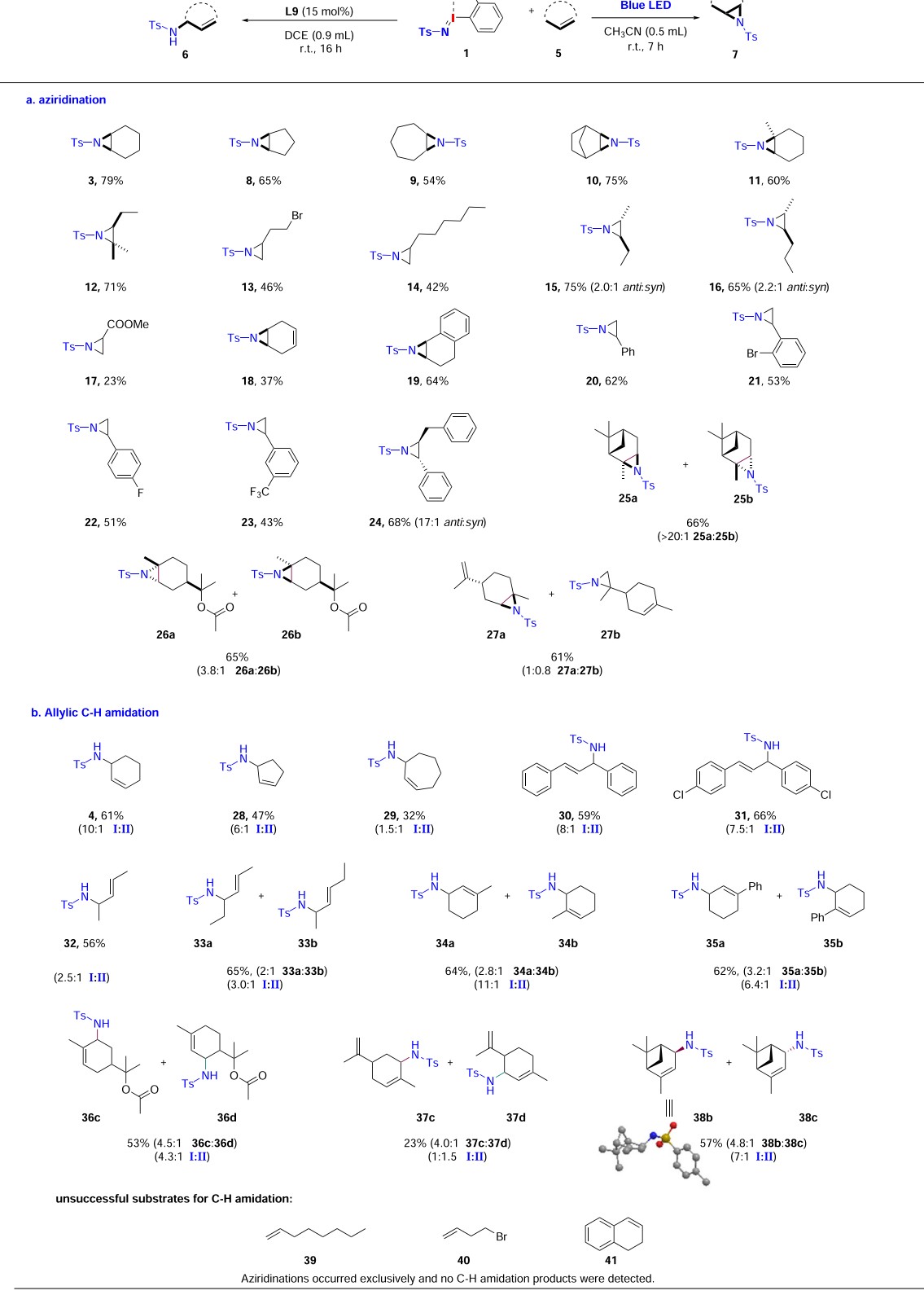

**Fig. 2 The reaction scope. a** The reaction scope of blue light-induced aziridination of alkenes. Reaction conditions: **1** (0.05 mmol), **5** (0.25 mmol), in CH₃CN (0.5 mL) under 12 W blue LEDs (435–445 nm) at room temperature for 7 h. Isolated yield. **b** The reaction scope of copper-catalyzed allylic C–H amination. Reaction conditions: **1** (0.1 mmol), **5** (0.5 mmol), [Cu(OTf)]₂·toluene (10 mol%), **L₉** (15 mol%), 4ÅMS (100 mg), DCE (0.9 mL), r.t., under N₂ for 16 h. Isolated yield. The ratio of isomers determined by ¹H NMR is given in the parenthesis. The ratio of C–H amination/C = C aziridination product (**I:II**) determined by ¹H NMR is given in the parenthesis.

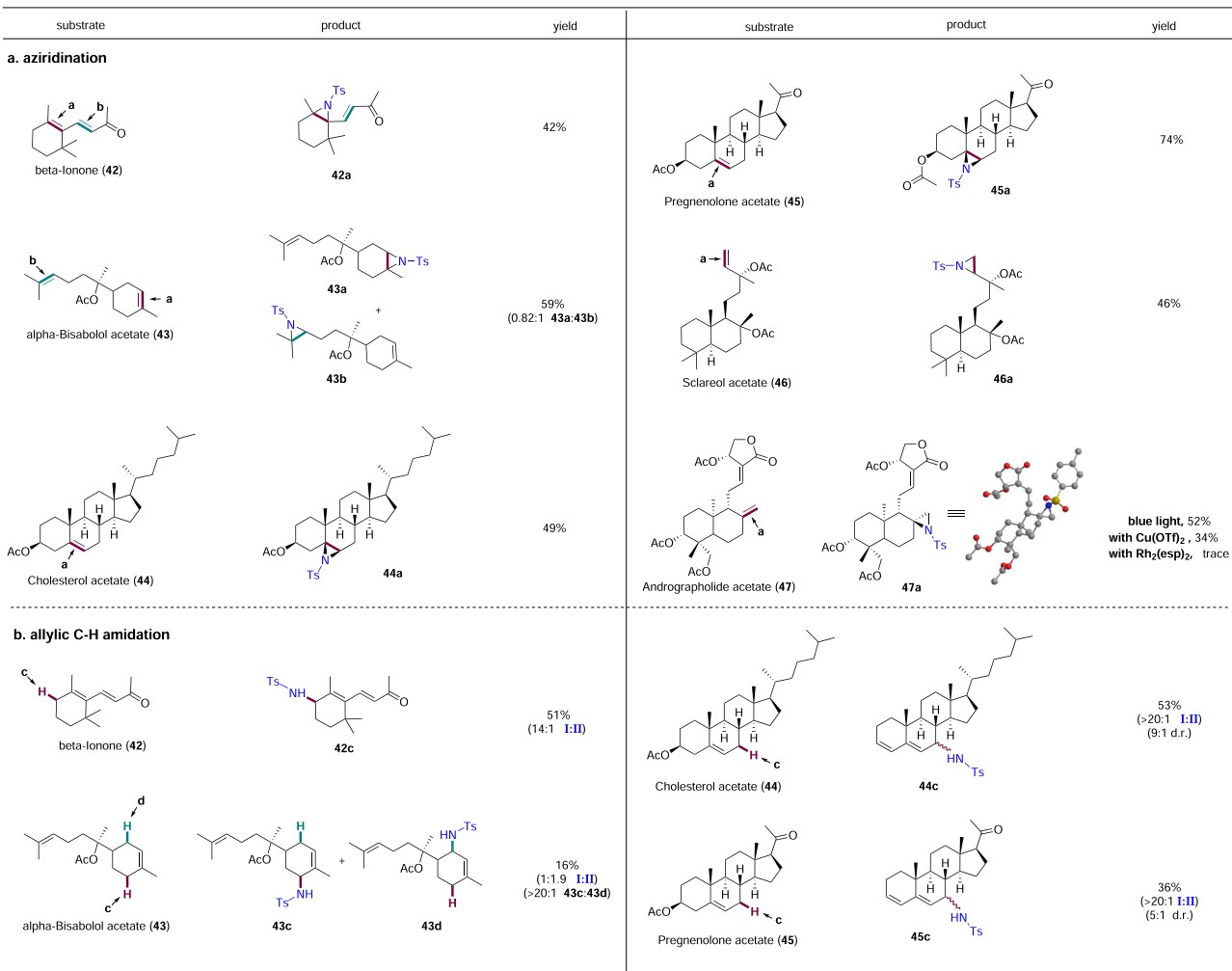

**Fig. 3 Synthetic application. a** Late-stage aziridination of biologically active natural products. Reaction conditions: **1** (0.05 mmol), natural products (0.1 mmol), in CH₃CN (0.5 mL) under 12 W blue LEDs (435–445 nm) at room temperature for 7 h. Isolated yield. **b** Late-stage allylic C–H amidation of biologically active products. Reaction conditions: **1** (0.1 mmol), natural products (0.2 mmol), [Cu(OTf)]₂·toluene (10 mol%), **L₉** (15 mol%), 4ÅMS (100 mg), DCE (0.9 mL), r.t., under N₂ for 16 h. Isolated yield. The ratio of isomers is given in the parenthesis. The ratio of C–H amidation/aziridination product (**I:II**) is given in the parenthesis.

C–H amination (Fig. 2b). Five- and six-membered rings selectively gave C–H amination products (**4** and **28**). However, reaction of cycloheptene showed less selectivity for C–H amination (1.5:1 **I:II**) (**29**), probably due to poor stabilization of an allylic radical, which were induced by competing transannular interactions. This provides indirect evidence that copper-catalyzed C–H amination may operate *via* a stepwise nitrene transfer pathway. The reaction of (*E*)-1,3-diarylpropenes and pent-2-ene gave the corresponding C–H amination products mainly and no isomerization was observed (**30–32**). (*E*)-hex-2-ene gave good selectivity for C–H amination and showed rearranged product, which also indicates a potential stepwise pathway (**33a** and **33b**). Good chemoselectivity for C–H amination was also noted for 1-methylcyclohexene and 1-phenylcyclohexene (**34–35**). Amination at the less hindered allylic C–H bonds is favored by a ratio of 2.8:1 and 3.2:1, respectively. Under copper catalysis, C–H amination of α-terpinylacetate occurred to deliver **36c** and **36d** in a 53% overall yield, where activation of less hindered C–H bond is preferred by a ratio of 4.5:1. The presence of terminal alkene in limonene results in poor selectivity for copper-catalyzed C–H amination, favoring less hindered C–H bond by a ratio of 4.0:1 (**37c** and **37d**). Pinene showed excellent chemoselectivity for C–H

amination (**38b and 38c**), providing the corresponding products in modest yields with *syn*-isomer (**38b**) as the major product. The absolute configuration of **38b** was determined by X-ray crystallography (see Supplementary Fig. 105 and Supplementary Data 1). In contrast, copper-catalyzed aziridination of terminal alkenes (**39** and **40**) and 1,2-dihydronaphthalene (**41**) is favored over C–H amination.

**Late-stage functionalization of biologically active natural products**. The developed C–H amination/C=C aziridination methods were applied to a diverse set of natural products and their derivatives, containing multiple alkene and allylic C–H functionalities. In most cases, chemo-site selectivity can be tuned by altering the reaction systems (photo-induction or copper catalysis) (Fig. 3). Aziridination of β-ionone **42** occurred exclusively on the cyclic C=C bond **a**, perhaps due to the electron poorness of C=C bond **b**, giving **42a** in 42% yield. Copper-catalyzed C–H amination of β-ionone **42** showed excellent chemo- and regio-selectivity, providing **42c** as the sole product in 51% yield. As expected, aziridination of α-bisabolol acetate **43** gave a 0.82:1 mixture of aziridine products (**43a** to **43b**) in 59% yield, with exo product preferred slightly over endo product. Switching to copper

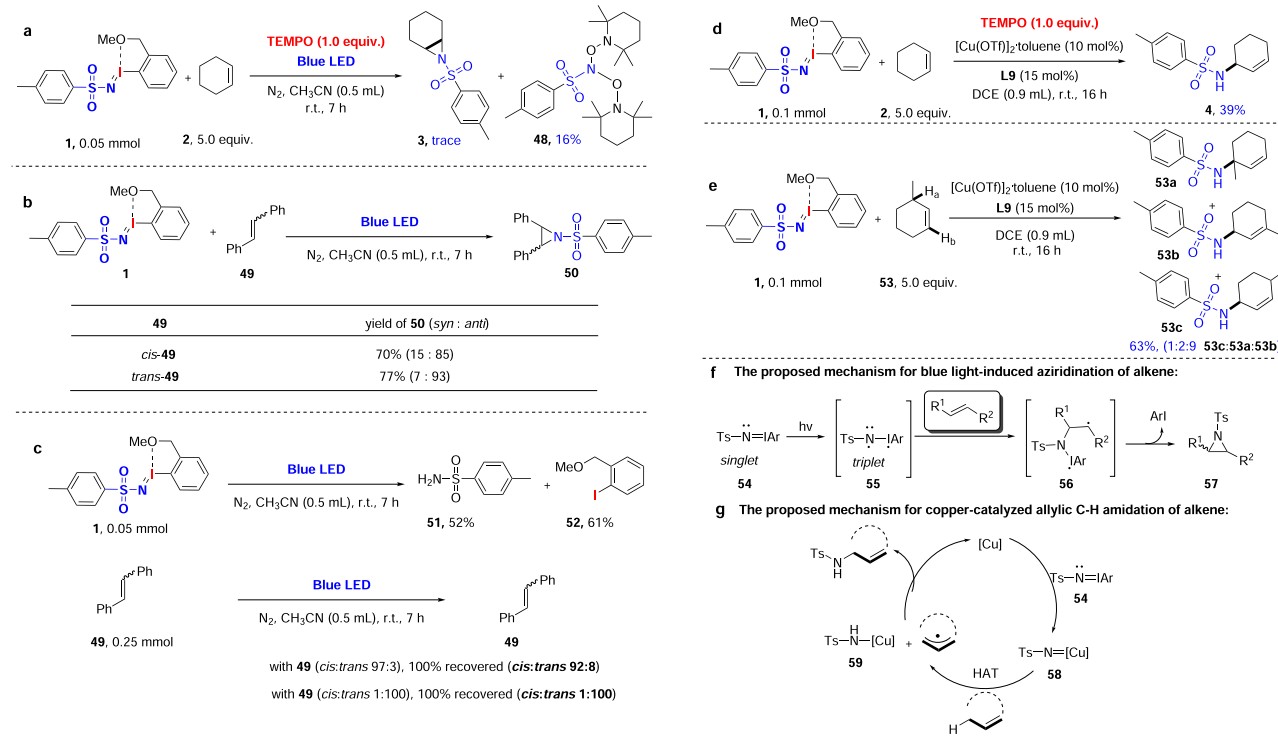

**Fig. 4 Mechanistic studies. a** Influence of the radical inhibitor TEMPO on the aziridination. **b** Study the aziridination of *cis-* or *trans-* alkenes. **c** Study the activation of substrates by blue LED irradiation. **d** Influence of TEMPO on the allylic C–H amination. **e** Allylic isomerization studies. **f, g** The proposed mechanisms.

catalysis resulted in site-selective C–H amination (**43c** and **43d**), but the aziridination process were also favored because of the presence of two electron-rich alkenes in **43**. However, site-selectivity for amination of the less hindered C–H bond is excellent with a ratio of >20:1. Both cholesterol acetate **44** and pregnenolone acetate **45** underwent facially selective aziridination with photo irradiation, leading to **44a** and **45a** as single diastereomers in acceptable yields, respectively. Pregnenolone acetate **45** presented higher aziridination efficiency (74% yield) than cholesterol acetate **44** (49% yield), probably due to the better solubility of pregnenolone acetate **45** in $CH_3CN$. Switching to copper catalysis, both cholesterol acetate and pregnenolone acetate gave C–H aminated diene products (**44c** and **45c**) in good chemo- and site-selectivity. The diene structure may result from elimination of acetate group in the presence of copper catalyst, which has been reported previously[67]. In the cases of sclareol and andrographolide acetate (**46** and **47**), aziridinations were observed exclusively at the terminal alkenes under both photo induction and copper catalysis (**46a** and **47a**), which is consistent with the reactivity of terminal alkenes in Fig. 2b. The absolute configuration of **47a** was characterized by *X-ray* diffraction (see Supplementary Fig. 106 and Supplementary Data 1). We also compared the aziridination efficiency between photo induction and metal catalysis. Surprisingly, photo induced aziridination of andrographolide acetate gave the best yield and selectivity. With copper catalysis, the aziridine product was obtained in 34% yield, along with some side products derived from copper-catalyzed ring-opening reaction of aziridine. In contrast, $Rh_2(esp)_2$ gave only trace amount of the desired aziridination product with most of the natural product intact.

**Mechanistic study**. To get a better understanding of the reaction mechanism, control experiments were carried out. As shown in Fig. 4a, subjecting **1** and **2** to aziridination conditions in the presence of (2,2,6,6-tetramethylpiperidinooxy) TEMPO as a radical inhibitor showed a substantial decrease in yield, which suggests the presence of radical intermediates. Additionally, compound **48**, which may derive from coupling of nitrene precursor with TEMPO, was obtained in 16% yield. Then we used *cis-* and *trans-* alkene to get stereochemical insight into the visible light-induced aziridination of alkenes (Fig. 4b). The photo-induced reaction of *cis-***49** gave the corresponding aziridine **50** in 70% overall yield with *syn/anti* ratio of 15:85, whereas *trans-***49** gave **50** in 77% yield with 7:93 *syn/anti* ratio. These results strongly suggest a two-step mechanism, which allows isomerization to proceed, rather than a concerted nitrogen atom transfer. Finally, in order to get insight into photo-induced activation of substrates, we investigated the conversion of iminoiodinane and alkene **49** under blue LED irradiation, separately. As shown in Fig. 4c, iminoiodinane was decomposed into 52% yield of sulfonamide **51** and 61% yield of **52** after irradiating for 7 h. In contrast, both the *cis-* and *trans-* isomers of **49** remained unchanged. These results suggest iminoiodinanes, rather than alkene substrates were activated under photo irradiation. Based on the studies above, a possible mechanism for visible light-induced aziridination was proposed as in Fig. 4f. The reaction started with photo-promoted conversion of iminoiodinane to an active intermediate **55** in the triplet state. The intermediate was rapidly trapped by alkene to furnish a radical intermediate **56**, which underwent intramolecular cyclization to provide the aziridine product **57**. For copper-catalyzed C–H amination, the addition of TEMPO led to a 25% decrease in yield (Fig. 4d). In addition, we note that copper-catalyzed C–H amination of 3-methylhexene generates a mixture of **53c**, **53a** and **53b** in a ratio of 1:2:9. A transposed double bond was observed in **53b**. All these results supported the presence of radical intermediates in the copper-catalyzed C–H amination. Base on this, a stepwise nitrene transfer pathway involving a copper-nitrene intermediate was proposed as in Fig. 4g.

In conclusion, we have developed visible light-induced aziridination and copper-catalyzed selective C–H amination. Both reaction systems demonstrate complementary chemoselectivities in alkenes, largely independent of substrate identities. Furthermore, combination of the above two methods allow for selective functionalization of biologically active natural products containing sites for both aziridination and C–H amination in nitrene transfer. A series of mechanistic studies indicate a two-step radical process was involved in both reaction systems.

## Methods

**General procedure for photo-induced aziridination of alkenes.** In a glovebox, a solution of **1** (0.05 mmol, 21 mg) in CH$_3$CN (0.5 mL) was added into a 2.0 mL vial. Then alkene (0.25 mmol of simple alkenes or 0.1 mmol of complex natural products) was added and the resulting mixture was stirred under 12 W blue LED (435–445 nm) at room temperature for 7 h. The resultant reaction mixture was purified by preparative TLC (hexane/ ethyl acetate 10:1 to 5:1).

**General procedure for copper-catalyzed allylic C–H amidation of alkenes.** In a glovebox, a solution of [Cu(OTf)]$_2$·toluene (10 mol%) and **L$_9$** (15 mol%) in DCE (0.9 mL) was added into a 2.0 mL vial. The mixture was stirred for 30 min. Then 4ÅMS (100 mg), alkene (0.5 mmol of simple alkenes or 0.2 mmol of complex natural products) and **1** (0.1 mmol) were added successively. The resulting mixture was stirred at room temperature for 16 h. The resultant reaction mixture was purified by preparative TLC (hexane/ethyl acetate 5:1).

## Data availability

The data supporting the findings of this study are available within the article and its Supplementary Information files. For experimental details and compound characterization data see Supplementary Methods and Supplementary Notes 1, 2. For $^1$H NMR and $^{13}$C NMR spectra see Supplementary Figs. 1–104. For the X-ray crystallographic data see Supplementary Fig. 105 for **38b**, Supplementary Fig. 106 for **47a** and Supplementary Data 1. The X-ray crystallographic coordinates for structures reported in this Article have been deposited at the Cambridge Crystallographic Data Centre (CCDC), under deposition number CCDC 2125791 for **38b** and 2125792 for **47a**. These data can be obtained free of charge from The Cambridge Crystallographic Data Centre via www.ccdc.cam.ac.uk/data_request/cif. Extra data are available from the corresponding authors upon request.

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

## Acknowledgements

This work was financially supported by National Natural Science Foundation of China (21977008, 21603246), Natural Science Foundation of Guangdong Province (2018A0303130052), and Shenzhen Basic Research Project (JCYJ20180503182116931).

## Author contributions

Q.X., C.C., and Z.Z. conceived, designed, and originated this project. Q.X., D.J., M.D., H.C., and J.Z. performed the experiments, obtained all spectroscopic data, and analyzed the results. Q.X., C.C., L.G., and Y.Z. co-wrote the manuscript. T.L. performed the biology experiments and data analysis. We would like to thank professor Zhen Yang (Peking University) for his support and valuable suggestions.

## Competing interests

The authors declare no competing interests.
