## [Peer Review File · Communications Chemistry]

Reviewers' comments:

Reviewer #1 (Remarks to the Author):

This manuscript describes a nice addition to intermolecular chemoselective nitrene transfer methods, where blue LEDs can be used to furnish aziridines, while Cu catalysis yields the allylic amine. Publication can be recommended after consideration of the following comments, some of which address the science and others the ease of presentation of the material.

Presentation style

- Figures and schemes: It would be better to use color schemes or arrows to highlight different sites/functional groups on substrates that could potentially undergo a reaction under each of the conditions, especially in Fig. 3. It is difficult to keep track of where a reaction occurs on a complex substrate when product structures are not consistently drawn, i.e., randomly flipped or rotated.
- There are a few minor spelling and grammatical errors.
- Table 1- it is a little difficult to make out the 'I' in structure 1. Perhaps use a different color?
- The major product in 24 is drawn as syn- should be drawn as anti.
- Figure 2- stereochemistry is not needed for 13-14, 17, 21-23.

Science

- What is the remainder of the mass balance for reactions in Table 2?
- Scope: Fig. 2b should include substrates that unsuccessfully undergo selective C-H amination and their selectivity data (e.g. terminal alkenes that are mentioned in the text without any structures and data to show). Scope in Fig. 3 is pretty sufficient, but discussions on the selectivity basis for certain interesting substrates could be more in-depth with the inclusion of structural models, computational data, etc.
- This work also suffers from having to use a large excess of substrate (5 equiv). Obviously, it is not attractive for late-stage functionalization to have to use a large excess. I see from the SI that the substrate/nitrene precursor ratios in Figure 3 are different than Figure 2. Could the authors comment on how this would be expected to impact overall yield? What is the remaining mass balance in the low-yielding reactions in Figure 3? Can the starting material be recovered?
- Figure 3 is pretty crowded and a lot of the yields are quite poor. Some of the substrates that are not all that complex (34-38) could be moved to the SI or commented on in the substrate scope (Figure 2).
- Why the differences in yield between 39a and 40a? The electronic effects should be remote.
- In Figure 5c, where do the two H on N come from? Are they abstracted from the MeCN solvent? What was the other by-product?
- Mechanistic studies: Might add radical clock experiments, although I do think it is clear from the experiments presented in the manuscript that the nitrene transfer is stepwise. Fig. 5e has typos in some of the structures

Curiosity-related questions

- I'm curious if you have any spectroscopic data on the o-methoxymethyl iminoiodinane? Optimizations of wavelength/light source? Given the claim that ortho-substituents render iminoiodinanes photoactive, does the reaction with PhI=NTs under visible light (Table 1, Entry 4) occur via the same pathway?
- Curious if iminoiodinanes derived from sulfamates or carbamates work in the chemistry?

- Table 2- any differences if the metal:ligand ratio is altered?
- I'm not sure the comparison of the SARS-CoV-2 activity is appropriate, given that the other two are approved drugs. The aziridine could be completely non-specific and therefore, this change might not be actually useful. I just wonder, as aziridines are not often found in drug-like molecules.

Reviewer #2 (Remarks to the Author):

The manuscript by Xing and co-workers reported the chemo-selective nitrene transfer for alkenes under two different reaction conditions: copper catalyst or catalyst-free visible-light conditions. Both allylic C–H amination and aziridination of alkenes was obtained in a tunable way. Authors further demonstrated the utilities of these methods to biologically active natural products. Importantly, the modified molecules have been shown much more activities than the parent molecules. Therefore, my feeling is that this manuscript is generally suitable to be published in communication chemistry. However, there are several very important points that the authors should address before its publication.

Questions and suggestions:

1. Copper complex sometimes can play a role as a photo-catalyst, so how about the reaction selectivity, if combining the copper catalyst and visible-light in the same reaction condition.
2. As author mentioned photo-induced aziridination of alkenes has been reported by Takemoto and co-workers (Ref 65). So, the comparable experiments between Takemoto's iminoindinanes and this iminoindinanes should be carried out under the copper catalyst condition.
3. In SI, the melt-point of the solid compounds is necessary. ¹H NMR spectra of compound 17, 18, 23, 24, 25, 42a, 42h, need to be purified again. ¹³C NMR spectra of 23, 42c didn't recognize the CF₃ group (SI: line 347-349 for 23; line 766-769).

Reviewer #3 (Remarks to the Author):

The authors report the development of a method for nitrene transfer based on the combination of visible light and copper catalysis.

The authors report two different methods from the same starting materials – an aziridination and a C–H amination.

Examples of both of these processes using the same type of iminoiodinane and other nitrate precursors are known in the literature (e.g., using Fe chem: *Green Chem.* 2021, 23, 9428; using Cu chem: *Org. Lett.* 2019, 21, 1926). The present process is novel but is certainly related to existing processes. The main novelty arises in the divergent reactivity when visible light is on or off, although this is also related to other processes. Having said this, I do believe that this is sufficiently novel to be publishable in this journal.

The manuscript follows the standard approach - hypothesis/design, optimisation, application. It is logically ordered and generally easy to follow.

However, there are issues that must be addressed before publication can be further considered:

1. The optimisation data presented is the same in the manuscript as in the SI. Very little additional detail is provided. I would suggest that additional data is provided to give a more full description of the variables assessed.

2. There are significant issues with the characterisation data. I sampled 15 of the compounds reported and found issues with the data. Some compounds have more protons/carbon than they should, such as 39b, (44 carbons reported for 36 expected, 59 protons vs. 55 expected), 42d, 42g (34 instead of 36). The SI needs thoroughly checked.

In principle, I believe this manuscript is publishable but it would be prudent to request a revision is reviewed before any further decision is made on publication.

May 2nd 2022

Manuscript number: COMMSCHEM-22-0025

Title: Combining visible-light induction and copper catalysis for chemo-selective nitrene transfer: a practical strategy for late-stage amination of natural products

[name redacted],

Thank you very much for your consideration in this work. We appreciate the assessment from you and reviewers. We would like to respond to comments from reviewers as below:

Referee 1:

Recommendation: Publish after minor revisions noted.

Referee Letter: This manuscript describes a nice addition to intermolecular chemoselective nitrene transfer methods, where blue LEDs can be used to furnish aziridines, while Cu catalysis yields the allylic amine. Publication can be recommended after consideration of the following comments, some of which address the science and others the ease of presentation of the material.

Presentation style

- **Figures and schemes:** It would be better to use color schemes or arrows to highlight different sites/functional groups on substrates that could potentially undergo a reaction under each of the conditions, especially in Fig. 3. It is difficult to keep track of where a reaction occurs on a complex substrate when product structures are not consistently drawn, i.e., randomly flipped or rotated.

Our response: Thank you very much for your valuable suggestion. In order to keep track of bond transformation under each of the conditions, we used color scheme, arrows and labels to highlight different sites on complex substrates. Figure 3 has been revised according to above suggestions.

Revised Figure 3:

substrate	product	yield	substrate	product	yield
a. aziridination					
		42%			74%
		59% (0.82:1 43a:43b)			46%
		49%			blue light, 52% with Cu(OTf) ₂ , 34% with Rh ₂ (esp) ₂ , trace
b. allylic C-H amidation					
		51% (14:1 I:II)			53% (>20:1 I:II) (8:1 d.r.)
		16% (1:1.9 I:II) (>20:1 43c:43d)			36% (>20:1 I:II) (5:1 d.r.)

Fig. 3 Synthetic application. **a** Late-stage aziridination of biologically active natural products. Reaction conditions: **1** (0.05 mmol), natural products (0.1 mmol), in CH₃CN (0.5 mL) under 12 W blue LEDs (435-445 nm) at room temperature for 7 h. Isolated yield. **b** Late-stage allylic C–H amidation of biologically active products. Reaction conditions: **1** (0.1 mmol), natural products (0.2 mmol), [Cu(OTf)₂·toluene (10 mol%), **L**₉ (15 mol%), 4ÅMS (100 mg), DCE (0.9 mL), r.t., under N₂ for 16 h. Isolated yield. The ratio of isomers given in the parenthesis. The ratio of C–H amidation/aziridination product (**I** : **II**) is given in the parenthesis.

- There are a few minor spelling and grammatical errors.

Our response: We feel sorry for our carelessness and great thanks for your point out. We have thoroughly reviewed our manuscript and corrected the spelling and grammatical errors:

1. Page 2, left column, line 74, “controled” was corrected to “controlled”,
2. Page 2, left column, line 86, “EuPA” was corrected to “Eupalmerin acetate”,
3. Page 2, left column, line 95, “olefines” was corrected to “olefins”,
4. Page 2, left column, line 105, “unstrated” was corrected to “unsaturated”,
5. Page 5, right column, line 291, “product” was corrected to “products”,
6. Page 6, left column, line 316, “induce-aziridination” was corrected to “induced aziridination”.

- Table 1- it is a little difficult to make out the 'I' in structure 1. Perhaps use a different color?

Our response: Thanks very much for this valuable advice. In order to make out the "I" in structure 1, we changed the font color of "I" to pink.

- The major product in 24 is drawn as syn- should be drawn as anti.
- Figure 2- stereochemistry is not needed for 13-14, 17, 21-23.

Our response: Thanks for your reminder. As shown in the revised Figure 2, the major product in **24** has been corrected to the *anti*-isomer. The stereochemistry for **13-14, 17, 21-23** has been omitted and the stereoisomers for other products have been revised carefully.

Revised Figure 2:

Fig. 2 The reaction scope. **a** The reaction scope of blue light-induced aziridination of alkenes. Reaction conditions: **1** (0.05 mmol), **5** (0.25 mmol), in CH₃CN (0.5 mL) under 12 W blue LEDs (435-445 nm) at room temperature for 7 h. Isolated yield. **b** The reaction scope of copper-catalyzed allylic C–H amidation. Reaction conditions: **1** (0.1 mmol), **5** (0.5 mmol), [Cu(OTf)₂·toluene (10 mol%), **L9** (15 mol%), 4ÅMS (100 mg), DCE (0.9 mL), r.t., under N₂ for 16 h. Isolated yield. The ratio of isomers determined by ¹H NMR is given in the parenthesis. The ratio of C–H amidation/C=C aziridination product (**I** : **II**) determined by ¹H NMR is given in the parenthesis.

Science

- What is the remainder of the mass balance for reactions in Table 2?

Our response: In Table 2, the ratio of iminoiodinane to alkene is 1:5. After reaction, alkene was recovered and *p*-toluenesulfonamide was obtained as the major side product, which was generated from decomposition of the iminoiodinane. Accordingly, the sentence “*p*-Toluenesulfonamide, which was generated from decomposition of the iminoiodinane, was obtained as the major side product.” was added to the footnote of Table 2.

- Figure 3 is pretty crowded and a lot of the yields are quite poor. Some of the substrates that are not all that complex (34-38) could be moved to the SI or commented on in the substrate scope (Figure 2).

Our response: Thanks for your valuable advice. As shown in revised Figure 2 and Figure 3, the substrates **34-36** in Figure 3 have been moved to Figure 2. Considering the structural similarity of **24** and **25**, we removed **25** from Figure 2a. On the other hand, the tosyl group was abbreviated as “Ts”, and thus more space was released. Accordingly, the compound No. were renewed as below:

Changes: **36a** to **25a**, **34a** to **26a**, **34b** to **26b**, **35a** to **27a**, **37b** to **27b**, **26** to **28**, **27** to **29**, **28** to **30**, **29** to **31**, **30** to **32**, **31a** to **33a**, **31b** to **33b**, **32a** to **34a**, **32b** to **34b**, **33a** to **35a**, **33b** to **35b**, **34c** to **36c**, **34d** to **36d**, **35c** to **37c**, **35d** to **37d**, **36b** to **38b**, **36c** to **38c**, **37** to **42**, **37a** to **42a**, **38** to **43**, **38a** to **43a**, **38b** to **43b**, **39** to **44**, **39a** to **44a**, **40** to **45**, **40a** to **45a**, **41** to **46**, **41a** to **46a**, **42** to **47**, **42a** to **47a**, **37b** to **42c**, **38c** to **43c**, **38d** to **43d**, **39b** to **44c**, **40b** to **45c**.

Revised Figure 2:

Fig. 2 The reaction scope. **a** The reaction scope of blue light-induced aziridination of alkenes. Reaction conditions: **1** (0.05 mmol), **5** (0.25 mmol), in CH₃CN (0.5 mL) under 12 W blue LEDs (435-445 nm) at room temperature for 7 h. Isolated yield. **b** The reaction scope of copper-catalyzed allylic C-H amidation. Reaction conditions: **1** (0.1 mmol), **5** (0.5 mmol), [Cu(OTf)₂·toluene (10 mol%), **L9** (15 mol%), 4ÅMS (100 mg), DCE (0.9 mL), r.t., under N₂ for 16 h. Isolated yield. The ratio of isomers determined by ¹H NMR is given in the parenthesis. The ratio of C-H amidation/C=C aziridination product (**I** : **II**) determined by ¹H NMR is given in the parenthesis.

Revised Figure 3:

substrate	product	yield	substrate	product	yield
a. aziridination					
		42%			74%
		59% (0.82:1 43a:43b)			46%
		49%			blue light, 52% with Cu(OTf) ₂ , 34% with Rh ₂ (esp) ₂ , trace
b. allylic C-H amidation					
		51% (14:1 I:II)			53% (>20:1 I:II) (8:1 d.r.)
		16% (1:1.9 I:II) (>20:1 43c:43d)			36% (>20:1 I:II) (5:1 d.r.)

Fig. 3 Synthetic application. **a** Late-stage aziridination of biologically active natural products. Reaction conditions: **1** (0.05 mmol), natural products (0.1 mmol), in CH₃CN (0.5 mL) under 12 W blue LEDs (435-445 nm) at room temperature for 7 h. Isolated yield. **b** Late-stage allylic C–H amidation of biologically active products. Reaction conditions: **1** (0.1 mmol), natural products (0.2 mmol), [Cu(OTf)₂·toluene (10 mol%), **L**₉ (15 mol%), 4ÅMS (100 mg), DCE (0.9 mL), r.t., under N₂ for 16 h. Isolated yield. The ratio of isomers is given in the parenthesis. The ratio of C–H amidation/aziridination product (**I**:**II**) is given in the parenthesis.

- **Scope:** Fig. 2b should include substrates that unsuccessfully undergo selective C–H amination and their selectivity data (e.g. terminal alkenes that are mentioned in the text without any structures and data to show). Scope in Fig. 3 is pretty sufficient, but discussions on the selectivity basis for certain interesting substrates could be more in-depth with the inclusion of structural models, computational data, etc.

Our response: Thanks for your advice. As shown in revised Figure 2b, structures of unsuccessful substrates (oct-1-ene **39**, 4-bromobut-1-ene **40** and 1,2-dihydronaphthalene **41**) for C–H amination and the corresponding description of reaction results have been added. Additionally, further discussions on the reactions of β-ionone **42**, α-bisabolol acetate **43**, Cholesterol acetate **44**, Pregnenolone acetate **45** and andrographolide acetate **47** have been added in the revised manuscript.

Changes:

1. Page 4, right column, lines 267 to 270, The sentence “In contrast, copper-catalyzed aziridination of terminal alkenes is favored over C–H amination.” was changed to “In contrast, copper-catalyzed aziridination of terminal alkenes (**39** and **40**) and 1,2-dihydronaphthalene (**41**) is favored over C–H amination.”
2. Page 5, left column, lines 285 to 287, The sentence “Aziridination of β -ionone **42** occurred exclusively on the cyclic C=C bond **a**, perhaps due to the electron poorness of C=C bond **b**, giving **42a** in 42% yield.” was added.
3. Page 5, right column, lines 290 to 293, The sentence “As expected, aziridination of α -bisabolol acetate **38** gave a 0.82:1 mixture of aziridine product (**38a** and **38b**) in 59% yield.” was changed to “As expected, aziridination of α -bisabolol acetate **43** gave a 0.82:1 mixture of aziridine products (**43a** to **43b**) in 59% yield, with endo product preferred slightly over exo product.”
4. Page 6, left column, lines 301 to 304, The sentence “Pregnenolone acetate **45** presented higher aziridination efficiency (74% yield) than cholesterol acetate **44** (49% yield), probably due to the better solubility of pregnenolone acetate **45** in CH_3CN .” was added.
5. Page 6, left column, lines 312 to 313, The sentence “which is consistent with the reactivity of terminal alkenes in Figure **2b**.” was added.

Revised Figure 2b:

• This work also suffers from having to use a large excess of substrate (5 equiv). Obviously, it is not attractive for late-stage functionalization to have to use a large excess. I see from the SI that the substrate/nitrene precursor ratios in Figure 3 are different than Figure 2. Could the authors

comment on how this would be expected to impact overall yield? What is the remaining mass balance in the low-yielding reactions in Figure 3? Can the starting material be recovered?

Our response: We conducted controlled experiments to clarify how the substrate/nitrene precursor ratios impact overall yield. As shown in Scheme 1 below, different alkene/nitrene ratios (2:1, 1:1, 1:2 and 1:5) led to different yields. Compared with the reaction with 1:1 substrate ratio, an excess of alkene or nitrene precursor leads to increased yields. Further increasing the alkene/nitrene precursor ratio results in slightly higher yields for both photo-induced aziridination and copper-catalyzed C-H amination. Considering the substrate availability, we use five equiv. of simple alkenes in Figure 2 and two equiv. of complex natural products in Figure 3. The reaction conditions were added in the footnotes of revised Figure 2 and Figure 3. The impacts of substrate/nitrene ratios on aziridination or C-H amination have been added in the revised Table S1 and Table S3, respectively. The corresponding descriptions were added in the revised manuscript.

Changes:

1. Page 3, right column, lines 166 to 169, Sentences “We also investigated the impact of substrate ratio and found that reducing the amount of alkene to one or two equivalent both led to a slight decrease in product yield (entries 9 and 10 in Table S1 of Supplementary Information). No improvement in yield was observed when the reaction time was prolonged to 12 h. (entry 11 in Table S1 of Supplementary Information).” were added.
2. Page 4, left column, lines 202 to 204, The sentence “Slightly lower yields were obtained when the amount of alkene was reduced to one or two equivalents, while the chemoselectivity was unaffected (entries 13 and 14 in Table S3 of Supplementary Information).” was added.

Take the amination of Cholesterol acetate **44** for example (**Scheme 2**), after aziridination or C–H amidation, the intact Cholesterol acetate **44** was recovered. *p*-Toluenesulfonamide, which was generated from decomposition of nitrene precursor, was the major side product.

Scheme 1. Impact of substrate ratio on overall yield.

Scheme 2. mass balance in amination of β -sitosterol acetate.

- Why the differences in yield between **39a** and **40a**? The electronic effects should be remote.

Our response: β -Sitosterol acetate (original No.: **39**) and pregnenolone acetate (original No.: **40**) are structurally similar and the electronic effects are remote. It's found that pregnenolone acetate (original No.: **40**), which is more polar than β -Sitosterol acetate (original No.: **39**), presented better solubility in CH_3CN . After photo-induced aziridination, the reaction mixture of **39** is soluble while much solid material was still observed in the reaction mixture of **40**. LC-MS characterization indicated that most of the solid materials are the starting material **39**. So, the difference in yield may result from the different solubility between β -Sitosterol acetate (original No.: **39**) and pregnenolone acetate (original No.: **40**) in CH_3CN .

Changes:

Page 6, left column, lines 301 to 304, The sentence “Pregnenolone acetate **45** presented higher aziridination efficiency (74% yield) than cholesterol acetate **44** (49% yield), probably due to the better solubility of pregnenolone acetate **45** in CH_3CN .” was added.

- In Figure 5c, where do the two H on N come from? Are they abstracted from the MeCN solvent? What was the other by-product?

Our response: As shown in revised Figure **4c** below, in the absence of alkene, photo irradiation induced decomposition of iminoiodinane. *p*-Toluenesulfonamide and 2-iodobenzyl ether were obtained in 52% and 61% yield, respectively. As for *p*-toluenesulfonamide, the two H on N may come from H_2O in the reaction system.

Changes:

Page 6, right column, lines 341 to 343, The sentence “As shown in Figure 4c, iminoiodinane was decomposed into 52% yield of sulfonamide 51 and 61% yield of 52 after irradiating for 7 h.” was added.

Revised Figure 4c. Decomposition of iminoiodinane under photo irradiation.

• **Mechanistic studies:** Might add radical clock experiments, although I do think it is clear from the experiments presented in the manuscript that the nitrene transfer is stepwise. Fig. 5e has typos in some of the structures

Our response: Thanks for your kind reminder. Control experiments in Revised Figure 4d, 4e and the rearranged product of (*E*)-hex-2-ene proved a stepwise pathway in copper-catalyzed C-H amidation. In order to further illustrate the reaction pathway in photo-induced aziridination, we synthesized (1-cyclopropylvinyl)benzene **2** as a molecular probe and investigated its aziridination under photo irradiation. As shown in **Scheme 3**, 4-methylbenzenesulfonamide was obtained as the major product in 47% yield. The corresponding aziridination product **4** or radical-induced cyclization product **5** was not detected. (1-Cyclopropylvinyl)benzene **2** was converted into a mixture of unknown compounds. However, control experiments in Revised Figure 4a and 4b support a stepwise nitrene transfer pathway in photo-induced aziridination.

As shown in the revised Figure 4e, typos have been corrected carefully.

Scheme 3. Radical clock experiment for photo-induced aziridination

Revised Figure 4e:

Curiosity-related questions

- I'm curious if you have any spectroscopic data on the *o*-methoxymethyl iminoiodinane? Optimizations of wavelength/light source? Given the claim that *ortho*-substituents render iminoiodinanes photoactive, does the reaction with PhI=NTs under visible light (Table 1, Entry 4) occur via the same pathway?

Our response: Thanks for your questions. We didn't measure the absorption spectra of *o*-methoxymethyl iminoiodinane. In 2018, Takemoto and coworkers measured the absorption spectra of *N*-trifluoroacetyl iminoiodinanes bearing a *ortho*-methoxymethyl group (*Angew. Chem. Int. Ed.* **2018**, *57*, 693-697). The maximum absorption peak was at 229 nm and they also observed an absorption at approximately 370 nm. Computational calculation indicated that the energy gap between the HOMO and LUMO corresponded to the 364 nm excitation wavelength. In 2019, Nemoto et al. demonstrated that the hypervalent iodine could also be activated with light from the apparently non-absorbing region via a direct S_0-T_n transition, which has been considered a forbidden process (Ref. 67: *Angew. Chem. Int. Ed.* **2020**, *59*, 6847-6852). Inspired by this, we envisioned that such *ortho*-substituted iminoiodinanes could be similarly activated under light of a longer wavelength, such as visible light, and acted as a nitrene precursor for the aziridination of alkenes. Besides blue LED, we also tried white LED as the light source and a comparable yield was observed (**Scheme 4**). The corresponding result has been added in the revised Table S1 (in Supplementary Information).

Change:

Page 3, right column, lines 163 to 166, The sentence "Besides blue LED, we also tried to use white LED as the light source and a comparable yield was observed (entry 3 in Table S1 of Supplementary Information)." was added.

Scheme 4. Photo-induced aziridination with different light sources.

The *ortho*-coordinating substituents improved the solubility and stability of iminoiodinane (*Angew. Chem. Int. Ed.* **2018**, *57*, 693-697). Furthermore, such coordinating groups would also stabilize the photo-excited state of the iminoiodinane, preventing generation of free nitrenes (*J. Org. Chem.* **2014**, *79*, 8977-8983), which can afford undesired products through non-selective reactions. The reaction with PhI=NTs under visible light may occur via the same way, but an obviously lower yield was obtained, probably due to its poorer solubility and the absence of

stabilization effect by ortho-substituents.

- Curious if iminoiodinanes derived from sulfamates or carbamates work in the chemistry?

Our response: Thanks for your question. We tried to synthesize but failed to isolate iminoiodinanes derived from sulfamate or carbamate.

- Table 2- any differences if the metal:ligand ratio is altered?

Our response: We conducted parallel experiments of copper-catalyzed C-H amidation with different metal:ligand ratios, from 1:1 to 1:2.5 (Table 1). Compared with the chemoselectivity in 1:1.5 metal/ligand ratio, the yield and chemoselectivity for C-H amination is poorer when the catalyst system was used in 1:1 metal/ligand ratio. Further increasing the metal/ligand ratio from 1:1.5 to 1:2 or 1:2.5 improved the overall yield to some extent, but failed to improve the chemoselectivity for C-H amination. These results have been added in Table S3 (in SI). The corresponding descriptions was added in the revised manuscript.

Change:

Page 4, left column, lines 192 to 202, The sentences “We also conducted parallel experiments of copper-catalyzed C–H amidation with different metal:ligand ratios, from 1:1 to 1:2.5. Compared with the chemoselectivity in 1:1.5 metal/ligand ratio, the yield and chemoselectivity for C–H amination is poorer when the catalyst system was used in 1:1 metal/ligand ratio (entry 10 in Table S3 of Supplementary Information). Further increasing the metal/ligand ratio from 1:1.5 to 1:2 or 1:2.5 improved the overall yield to some extent, but failed to improve the chemoselectivity for C–H amination (entries 11 and 12 in Table S3 of Supplementary Information).” were added.

Table 1. Optimization of the metal/ligand ratios for copper-catalyzed C–H amidation ^{a,b,c,d} .			
			
Entry	Metal/Ligand	Yield ^b (%)	3/4
1	1:1	57	1/5
2	1:1.5	67	1/10
3	1:2	74	1/7.1
4	1:2.5	79	1/8.3
^a Reaction conditions: 1 (0.1 mmol), 2 (0.5 mmol), [Cu(OTf) ₂]·toluene (10 mol%), L9 (10–25 mol%), r.t., in solvent (0.9 mL) under N ₂ for 16 h. ^b Isolated yield. ^c 4ÅMS (100 mg) was added. ^d p-Toluenesulfonamide, which was generated from decomposition of the iminoiodinane, was obtained as			

the major side product.

- I'm not sure the comparison of the SARS-CoV-2 activity is appropriate, given that the other two are approved drugs. The aziridine could be completely non-specific and therefore, this change might not be actually useful. I just wonder, as aziridines are not often found in drug-like molecules.

Our response: Thanks for your kind comment. Andrographolide is commonly used as traditional Chinese medicine and has effects of anti-inflammatory, antibacterial, antiviral, antitumor and immune regulation. We have been dedicated to modifying andrographolide and finding promising hit compounds with strong antiviral activity against SARS-CoV-2. Herein, chloroquine and remdesivir were used as positive controls to evaluate the anti-SARS-CoV-2 activity of andrographolide derivatives. The aziridine derivatives did exhibit good inhibition activity toward SARS-CoV-2, but aziridines are rarely found in drug-like molecules. In view of this, remodeling of the aziridine products and the corresponding bioactivity studies (original Figure 4) were omitted from the revised manuscript.

Reviewer #2:

The manuscript by Xing and co-workers reported the chemo-selective nitrene transfer for alkenes under two different reaction conditions: copper catalyst or catalyst-free visible-light conditions. Both allylic C–H amination and aziridination of alkenes was obtained in a tunable way. Authors further demonstrated the utilities of these method to biologically active natural products. Importantly, the modified molecules have been shown much more activities than the parent molecules. Therefore, my feeling is that this manuscript is generally suitable to be published in communication chemistry. However, there are several very important points that the authors should address before its publication.

Questions and suggestions:

1. Copper complex sometimes can play a role as a photo-catalyst, so how about the reaction selectivity, if combining the copper catalyst and visible-light in the same reaction condition.

Our response: Thanks for your kind advice. We conducted the reaction with copper catalysis under photo irradiation. As shown in Scheme 5 below, C-H amination was preferred in DCE to give the corresponding amination product **4** in 55% yield, along with a 1:5.2 ratio of **3/4**. In contrast, the reaction in CH₃CN resulted in much lower yield for C-H amination and poorer chemoselectivity (1:1.7). In general, combining the copper catalyst and visible light in the same reaction condition led to lower yield and poorer chemoselectivity compared with the experiments under separate conditions.

Scheme 5. Copper-catalyzed amination of cyclohexene under blue LED irradiation

Entry	Solvent	yield of 3 (%)	yield of 4 (%)	ratio of 3 to 4 (3/4)
1	CH ₃ CN	14	24	1:1.7
2	DCE	11	55	1:5.2

2. As author mentioned photo-induced aziridination of alkenes has been reported by Takemoto and co-workers (Ref 65). So, the comparable experiments between Takemoto's iminoindinanes and this iminoindinanes should be carried out under the copper catalyst condition.

Our response: Thanks for your advice. According to the reactivity of Takemoto's iminoiodinanes, we prepared a series of iminoiodinanes and compared their activities in copper-catalyzed C-H amination (**Scheme 6**). In general, the corresponding C-H amination products were successfully obtained in moderate yields using **1a** or **1b** as nitrogen sources, respectively. Both yield and chemoselectivity for C-H amination with **1b** was slightly lower than those with **1a**. Iminoiodinane **1c** gave trace amounts of amination and aziridination products, probably due to its poor solubility in DCE. The experiments above have been added in Supplementary Information as **Scheme S1**. The corresponding descriptions were added in the revised manuscript.

Changes:

Page 4, left column, lines 205 to 218, The sentences "According to the reactivity of iminoiodinanes reported by Takemoto and coworkers, we prepared another two iminoiodinanes and compared their activities in copper-catalyzed C-H amination (**Scheme S1** in Supplementary Information). In general, the corresponding C-H amination product was successfully obtained in moderate yields using *ortho*-methoxymethyl or *ortho*-nitro iminoiodinanes of *p*-toluenesulfonamide as nitrogen sources, respectively. Both yield and chemoselectivity for C-H amination with *ortho*-methoxymethyl iminoiodinane was slightly higher than that with *ortho*-nitro iminoiodinane. *ortho*-Nitro iminoiodinane of *ortho*-nitrosulfonamide gave only trace amounts of C-H amination and aziridination products, probably due to its poor solubility in DCE." were added.

Scheme 6. Copper-catalyzed amination of cyclohexene with different iminoiodinanes

3. In SI, the melt-point of the solid compounds is necessary. ¹H NMR spectra of compound 17, 18, 23, 24, 25, 42a, 42h, need to purified again. ¹³C NMR spectra of 23, 42c didn't recognize the CF₃ group (SI:line 347-349 for 23; line 766-769).

Our response: Thanks for your kind reminder. The melt-point of solid products have been determined and added in SI. ¹H NMR and ¹³C NMR spectra of compound 17, 18, 23 and 42a (which has been renewed to 47a) have been purified and renewed in SI. Now the CF₃ group in the ¹³C NMR spectra of 23 can be recognized clearly. As shown in the revised manuscript, remodeling of the aziridine products and the corresponding bioactivity studies (Figure 4) were omitted because aziridines are rarely found in drug-like molecules. Accordingly, some andrographolide derivatives including 42b, 42c, 42d, 42e, 42f, 42g, 42h, 42i, 42j and 42k were removed from both the revised manuscript and SI. And considering the structural similarity of 24 and 25, we removed 25 from Figure 2a. The corresponding spectra of 25 and 42b-k were removed from SI.

Purified NMR spectra of 17, 18, 23 and 42a (which has been renewed to 47a):

¹H NMR spectra for compound 17

¹³C NMR spectra for compound 17

¹H NMR spectra for compound 18

¹³C NMR spectra for compound 18

¹H NMR spectra for compound 23

¹³C NMR spectra for compound 23

¹H NMR spectra for compound 42a (which has been renewed to 47a)

¹³C NMR spectra for compound 42a (which has been renewed to 47a)

Reviewer #3:

The authors report the development of a method for nitrene transfer based on the combination of visible light and copper catalysis.

The authors report two different methods from the same starting materials – an aziridination and a C–H amination.

Examples of both of these processes using the same type of iminoiodinane and other nitrate precursors are known in the literature (e.g., using Fe chem: *Green Chem.* 2021, 23, 9428; using Cu chem: *Org. Lett.* 2019, 21, 1926). The present process is novel but is certainly related to existing processes. The main novelty arises in the divergent reactivity when visible light is on or off, although this is also related to other processes. Having said this, I do believe that this is sufficiently novel to be publishable in this journal.

The manuscript follows the standard approach - hypothesis/design, optimisation, application. It is logically ordered and generally easy to follow.

In principle, I believe this manuscript is publishable but it would be prudent to request a revision is reviewed before any further decision is made on publication.

However, there are issues that must be addressed before publication can be further considered:

1. The optimisation data presented is the same in the manuscript as in the SI. Very little additional detail is provided. I would suggest that additional data is provided to give a more full description of the variables assessed.

Our response: Thanks for your valuable suggestion. As for photo-induced aziridination, we also investigated the impact of other parameters including different light source, substrate ratio and reaction time. The corresponding results have been added in the **revised Table S1**. Besides the optimization data presented in Table 2, influence of metal/ligand ratio as well as substrate ratio were also studied. The details of data are provided in the **revised Table S3** as below.

Revised Table S1. Optimization of the reaction conditions for photo-induced aziridination of alkenes.^{a,b}

Entry	Change from the “standard conditions”	Yield [%]
1	None	79
2	In the dark	no reaction
3	white LED instead of blue LED	75
4	In air instead of N ₂	72
5	PhI=NTs instead of 1	66
6	DCE instead of CH ₃ CN	53
7	THF instead of CH ₃ CN	not detected
8	DMF instead of CH ₃ CN	not detected
9	1.0 equiv. of 2	66
10	2.0 equiv. of 2	72
11	prolong reaction time to 12 h	77

^aReaction conditions: **1** (0.05 mmol), **2** (0.25 mmol), in CH₃CN (0.5 mL) under 12 W blue LEDs (435-445 nm) at room temperature for 7 h. ^bIsolated yield.

Revised Table S3. Optimization of the reaction conditions for copper-catalyzed allylic C–H amidation---screening of solvents, metal/ligand ratios and alkene/iminoiodinane ratios. ^{a,b}

Entry	Solvent	Ligand (mol %)	Yield (%)		3aa/4aa
			3	4	
1	CH ₃ CN	L ₇ (15 mol%)	3	11	1/4
2	DCE	L ₇ (15 mol%)	4	30	1/7.5
4	DMSO	L ₇ (15 mol%)	n.d.	n.d.	-----

5	DMA	L ₇ (15 mol%)	trace	trace	-----
6	MeOH	L ₇ (15 mol%)	n.d.	n.d.	-----
8	NMP	L ₇ (15 mol%)	trace	trace	-----
10	DCE	L ₈ (15 mol%)	4	34	1/8.5
9	DCE	L₉ (15 mol%)	4	44	1/11
10^c	DCE	L₉ (15 mol%)	6	61	1/10
11^c	DCE	L₉ (10 mol%)	10	47	1/5
12^c	DCE	L₉ (20 mol%)	9	65	1/7.1
13^c	DCE	L₉ (25 mol%)	9	70	1/8.3
14^{c,d}	DCE	L₉ (15 mol%)	5	54	1/10
15^{c,e}	DCE	L₉ (15 mol%)	5	57	1/11

^aReaction conditions: **1** (0.1 mmol), **2** (0.5 mmol, 5 equiv.), [Cu(OTf)₂·toluene (10 mol%), **L** (15 mol%), r.t., in solvent (0.9 mL) under N₂ for 16 h. ^bIsolated yield. ^c4ÅMS (100 mg) was added. ^d 1.0 equiv. of **2** was used. ^e 2.0 equiv. of **2** was used.

2. There are significant issues with the characterisation data. I sampled 15 of the compounds reported and found issues with the data. Some compounds have more protons/carbon than they should, such as **39b**, (44 carbons reported for 36 expected, 59 protons vs. 55 expected), **42d**, **42g** (34 instead of 36). The SI needs thoroughly checked.

Our response: Thanks very much for your kind reminder. We checked all the spectra of products in SI thoroughly. ¹H NMR and ¹³C NMR spectra of compound **17**, **18**, **23** and **42a** (which has been renewed to **47a**) have been purified and renewed in SI. The original **39b** was the aziridination product of β-sitosterol acetate (the original **39**). It's hard for us to purchase more β-sitosterol or its acetate derivative because of the epidemic. Cholesterol acetate (numbered as **44** in the revised Figure 3), which is

structurally similar to β -sitosterol acetate, was available. So, we used Cholesterol acetate instead of β -sitosterol acetate as the substrate for aziridination and C-H amination. The corresponding products **44a** and **44c** were isolated and the purified NMR spectra, HRMS data as well as melting point were provided in SI. As a result, the data for the original **39b** was removed from the revised manuscript and SI. As shown in the revised manuscript, remodeling of the aziridine products and the corresponding bioactivity studies (original Figure 4) were omitted because aziridines are rarely found in drug-like molecules. Accordingly, some andrographolide derivatives including **42b**, **42c**, **42d**, **42e**, **42f**, **42g**, **42h**, **42i**, **42j** and **42k** were removed from both the revised manuscript and SI. And considering the structural similarity of **24** and **25**, we removed **25** from Figure 2a. The corresponding spectra of **25** and **42b-k** were removed from SI.

^1H NMR and ^{13}C NMR spectra of compound **17**, **18**, **23**, **42a** (which has been renewed to **47a**), **44a** and **44c** have been purified and renewed in SI.

Purified NMR spectra of **17**, **18**, **23**, **42a** (which has been renewed to **47a**), **44a** and **44c**:

^1H NMR spectra for compound **17**

¹³C NMR spectra for compound 17

¹H NMR spectra for compound 18

¹³C NMR spectra for compound 18

¹H NMR spectra for compound 23

¹³C NMR spectra for compound 23

¹H NMR spectra for compound 42a (which has been renewed to 47a)

¹³C NMR spectra for compound 42a (which has been renewed to 47a)

¹H NMR spectra for compound 44a

¹³C NMR spectra for compound 44a

¹H NMR spectra for compound 44c

¹³C NMR spectra for compound 44c

With these changes, we hope that our revised manuscript now could be reconsidered to be accepted by *Communications Chemistry*.

Thank you for your favorable consideration.

Yours sincerely,
 Zhendong Zhu, Ph. D.
 Research Fellow
 BayRay Innovation Center
 Shenzhen Bay Laboratory
 Shenzhen 518132, China
 E-mail: zhuzd@szbl.ac.cn

REVIEWERS' COMMENTS:

Reviewer #1 (Remarks to the Author):

The authors have carefully considered all comments from the reviewers and have responded satisfactorily. I am happy to support publication.

Reviewer #2 (Remarks to the Author):

After consideration of author's revision, I recommended that this work could be published under current stage.

Reviewer #3 (Remarks to the Author):

The authors have attended to the requested revisions in full and I believe the study can now be considered for publication. I thank the authors for enthusiastically engaging with the review process.